# Effects of Different Isometric Training Programs on Muscle Size and Function in the Elbow Flexors

**DOI:** 10.3390/ijerph20053837

**Published:** 2023-02-21

**Authors:** ZhenFei Zou, Naoshi Morimoto, Miyuki Nakatani, Hirotsugu Morinaga, Yohei Takai

**Affiliations:** 1National Institute of Fitness and Sports in Kanoya, Kanoya 8912393, Kagoshima, Japan; 2Joint Master’s Program in International Development and Peace through Sport, University of Tsukuba, 1-1-1 Tennoudai, Tsukuba 3058577, Ibaraki, Japan; 3The Center for Liberal Arts, Meiji Gakuin University, 1518 Kamikurata, Totsuka, Yokohama 2448539, Kanagawa, Japan

**Keywords:** muscle thickness, MVC, ultrasound, resistance training

## Abstract

It remains unknown whether a regimen of a combination of high- and low-intensity resistance training increases muscle size and maximal voluntary isometric contraction (MVC) simultaneously. This study aimed to clarify the effect of the combination of high- and low-intensity resistance training on muscle size and neuromuscular function in the elbow flexors. Sixteen male adults participated in a 9-week isometric training regimen in elbow joint flexion of each arm. We randomly assigned two different training regimens to left and right arms: one aiming to strengthen maximal strength (ST) and the other aiming to develop muscle size as well as maximal strength, which consists of one contraction to volitional failure with 50% of MVC added to ST (COMB). Following the 3-week training to volitional failure as familiarization, the participants conducted the 6-week ST and COMB training in each arm. Before the intervention, and at the third (Mid) and ninth (Post) weeks, MVC and muscle thickness in the anterior part of the upper arm (ultrasound) were measured. Muscle cross-sectional area (mCSA) was derived from the obtained muscle thickness. From Mid to Post, the relative change in MVC was similar in both arms. The COMB regimen increased muscle size, but no significant change was found in ST. Following the 3-week isometric training to volitional failure, the 6-week training regimen for developing maximal voluntary and muscle hypertrophy increased MVC, with increasing mCSA, and the training-induced change in MVC was similar to that for developing maximal voluntary strength alone.

## 1. Introduction

Resistance training with isometric and dynamic contraction is an effective modality for developing force-generating capacity, such as maximal voluntary isometric contraction (MVC) and one repetition maximum (1RM) [1,2]. Many earlier studies have demonstrated that an increase in MVC depends not only on muscle hypertrophy, but also on changes in neural factors [3], such as improvement of coordination or learning [4] and increasing agonist muscle activity [5,6,7]. Regimens in resistance training have been classified into two major types: one aiming to increase force-generating capacity without muscle hypertrophy, and the other aiming to increase force-generating capacity and to increase muscle size. The former consists of high-intensity resistance exercises with low repetitions (>90% of 1RM and 3–5 repetitions) (ST-type), the latter of moderate-intensity resistance exercises with high repetitions (60–80% of 1RM and 8–12 repetitions) (HYP-type). The ST-type training has been shown to improve force-generating capacity due to neural adaptation, with less effect on muscle size [8,9]. On the other hand, the HYP-type training has been shown to increase force-generating capacity as well as muscle size [9,10]. Following 10–20 weeks of HYP-type training, an increase in muscle cross-sectional area (mCSA) is negatively related to changes in muscular activity and voluntary activation in the knee extensors [9], indicating that one or the other may predominantly adapt in the adaptation to resistance training for muscle hypertrophy and neural activation.

For optimizing force-generating capacity, a periodized training program has been shown to be more effective, compared to a non-periodized program, and then it has been recommended that HYP-type training is followed by ST-type training [11,12]. However, the HYP-type training regimen has the problem of taking a long time to achieve strength gain. As one of the ways to solve this problem, Goto et al. [8] have suggested that, following resistance training for muscle hypertrophy, a combination of high- and low-intensity resistance exercises is effective for simultaneous strength gain and muscle hypertrophy. In an earlier study [8], however, the training-induced change in muscle size was not significant, although the training regimen increased muscular strength.

Force-generating capacity is proportional to muscle size [13,14]. Based on the corresponding relationship, strength gain without muscle hypertrophy leads to an increase in the force-generating capacity relative to muscle size [8,15], known as muscle quality [16]. On the other hand, strength gain with muscle hypertrophy does not alter muscle quality [14]. In the earlier study [8], the increase in MVC/mCSA was the same between the ST-type training and the combination of high- and low-intensity resistance exercises. This finding implies that ST-type, and the combination of high- and low-intensity resistance training, do not differ in their adaptation to muscle quality. Given that the combination of high- and low-intensity resistance training causes the abovementioned adaptation, the adaptation to MVC/mCSA would be lower in the combination of high- and low-intensity resistance training than in ST-type training. Therefore, it is necessary to examine whether or not the combination of high- and low-intensity resistance training induces strength gain with muscle hypertrophy. This study aimed to clarify the effect of the combination of high- and low-intensity resistance training on muscle size and neuromuscular function in the elbow flexors. We tested a hypothesis that a combination of high- and low-intensity resistance training would induce force-generating capacity with increasing muscle size, but the training induced change in muscle quality would be lower, compared to ST-type training. In this study, we adopted the modality of isometric training, because isometric contractions are a highly reliable means of assessing and tracking changes in force production [1].

## 2. Materials and Methods

### 2.1. Participants

Male adults (N = 16; age = 22 ± 3 years, height = 174 ± 7 cm, weight = 76 ± 14 kg) participated in this study. None of the participants reported illnesses or taking prescribed medications for cardiovascular, metabolic, or orthopedic disorders. Prior to the following intervention, we randomly assigned two different training regimens to left and right arms: one aiming to strengthen maximal strength (ST) and the other aiming to develop muscle size as well as maximal strength (COMB). We explained the purpose and methods of the study in detail and obtained the participants’ informed consent before they took part in the experiment. We conducted this study with the approval of the National Institute of Fitness and Sports, in Kanoya (No. 11−102).

### 2.2. Experimental Approach to the Problem

Participants conducted a 9-week isometric training program (3 sessions/week) in elbow joint flexion, with a 90 degree flex of both arms (Table 1). For the first 3 weeks (familiarization period), participants conducted the same training program in both arms. For the remaining 6 weeks (experimental period), two different training programs were prescribed for each arm. For the familiarization period, the participants sustained elbow flexion force until volitional failure (i.e., inability of the subject to maintain the target force for more than 3 seconds consecutively, despite strong verbal encouragements [17]) at a given intensity per set, and then conducted three sets, with intervals of 3 min, in a training session. Training intensity was set at 60–80% of MVC, and increased weekly (two sessions at 60% MVC, three sessions at 70% MVC and three sessions at 80% MVC). The total number of training sessions being eight. The training frequency was three times a week. For the experimental period, one arm exerted a 3 s elbow flexion force five times per set, to exceed 90% MVC, with an interval of 5 s. The other arm sustained elbow flexion force until volitional failure at 50% MVC, after performing the same program as the abovementioned training. If the participants could not exert the elbow flexion force exceeding 90% MVC, we asked the participants to exert as much force as possible for 3 s. The number of sets was 3–4 in a session, and the total number of training sessions was 18.

According to the methods reported by a previous study [18], MVC and muscle size were measured at the beginning of the week to adjust the training intensity for each participant. As indices of training volume, we recorded the exerted force in every training session, and the computed impulse and mean elbow flexion force exerted in a session were calculated from the obtained data (see below in MVC). The impulse was summed up, and the exerted force was averaged in a week.

### 2.3. Muscle Thickness in Anterior Upper Arm

As a variable representing the size of elbow flexors, muscle thickness in the anterior part of the upper arm (MT) was measured by an ultrasonograph (ProSound Alpha6, Hitachi Aloka Medical, Japan), with a linear-array probe (7.27 MHz). During the measurement, the participants stood upright with their arms relaxed and extended. Following an earlier study [19], the MT was obtained at 60% of the upper arm length (the distance from the acromial process to the lateral epicondyle of the humerus). The scanning head, together with water-soluble transmission gel, which provided acoustic contact without depression of the skin, was placed perpendicular to the tissue interface at each of the marked sites. Distortion of tissue due to excessive compression was eliminated by ensuring that no movement of tissues occurred in the real-time ultrasonic image. The MT was defined as the distance from the subcutaneous adipose tissue-muscle interface, to the muscle-bone interface. All images were analyzed with image analysis software (Image J ver. 1.47, NIH, USA). The muscle cross-sectional area index (mCSA) was computed from the following equation [20]:mCSA = π × (MT/2)^2^
where π is a constant, 3.14, and MT is in cm.

### 2.4. Maximal Voluntary Isometric Contraction (MVC)

MVC in elbow joint flexion was measured by using a custom-made myometer, with tension/compression load cells (TR22S, SOHGOH KEISO CO., LTD., Japan). The elbow joint was held at a 90° flexed position (0° corresponds to full elbow extension). Participants were seated on an adjustable chair with the shoulder, and hip joints flexed at 90°. The force signals were amplified and attenuated with a low-pass filter (<100 Hz, DPM-912B, KYOWA, Japan). The force and angle signals were sampled at a frequency of 2 kHz, via a 16-bit analog/digital converter (PowerLab/16s: ADInstruments Sydney, Australia), and stored on a personal computer. The highest value among the two or three isometric forces was adapted as the elbow flexion MVC force. The value was divided by mCSA (MVC/mCSA) as an index of muscle quality.

### 2.5. Statistical Analysis

Descriptive data were presented as mean ± SD. The independent variables were mCSA, MVC, and MVC/mCSA. To test the homoscedasticity of within-subjects factor, a Mauchly’s sphericity test was applied, and the Greenhouse–Geisser statistic was used when the sphericity assumption was violated. For the familiarization and experimental periods, a two-way repeated measures analysis of variance (ANOVA: 2 groups × 2 time) was used to test the main effects of group and time, and their interaction on the measured variables, respectively. When appropriate, a simple main effect test was used to test the significance of the groups’ difference, for post hoc comparison.

We assessed the data using magnitude-based inferences for practical significance [21]. We used qualitative inferences to assess the differences in the independent variables between Mid (the third training week) and each measurement point, using a modified statistical spreadsheet [22]. Effect size (ES) and 90% confidence limits (CL) were calculated as threshold values: ≤0.2 (trivial), >0.2 (small), >0.6 (moderate), >1.2 (large), >2.0 (very large), and >4.0 (extremely large). We also assessed the qualitative change in higher or lower independent variables compared to Mid: <1% almost certainly not, 1–5% very unlikely, 5–25% unlikely, 25–75% possible, 75–95% likely, 95–99% very likely, and >99% almost certainly. If the chance of higher or lower differences was >5%, then the true difference was assessed as unclear.

The Pearson’s product-moment correlation coefficient (r) was calculated, to examine the relationships between relative changes in the MVC/mCSA ratio and its value at the beginning of the week. The magnitude of r was assessed with the following thresholds: < 0.1 (trivial), 0.1–0.3 (small), 0.3–0.5 (moderate), 0.5–0.7 (large), 0.7–0.9 (very large), and 0.9–1.0 (almost perfect) [22]. The correlation magnitude was deemed unclear if 90% CL overlapped small positive and negative values [21]. We computed the *p*-value using statistical software (IBM SPSS Statistics 25, IBM, Japan). The level of significance was set at 5%.

## 3. Results

### 3.1. Familiarization Period

No significant group-related differences in impulse and mean elbow flexion were found (Figure 1). The ANOVAs revealed a significant main effect of time, and no significant interaction between group and time in mCSA, MVC, and MVC/mCSA (Figure 2), indicating that the 3-week isometric training to volitional failure increased the corresponding variables to the same extent in both arms.

### 3.2. Experimental Period

Total of impulse summed weekly was greater in COMB than ST (F = 192.369, η_p_ = 0.865, *p* < 0.001) through the experimental period (Figure 1). Mean elbow flexion force exerted in training sessions tended to be higher in ST than COMB, but the level of significance was not reached (*p* = 0.073).

There was a significant interaction between group and time in MT (F = 12.328, η_p_ = 0.291, *p* = 0.001) and mCSA (F = 11.248, η_p_ = 0.273, *p* = 0.002). As the results of a test of the simple main effect, MT and mCSA in COMB were significantly increased (very likely with small), but no significant changes in the corresponding variables were found in ST (very unlikely with trivial) (Figure 2). For MVC, no significant interaction between group and time was found (F = 0.268, η_p_ = 0.009, *p* = 0.608), and a main effect of time was significant (F = 116.082, η_p_ = 0.795, *p* < 0.001) (Figure 2). Both types of training regimens increased MVC to the same extent in ST and COMB (almost certain with large). For MVC/mCSA, there was a significant interaction between group and time (F = 6.621, η_p_ = 0.181, *p* = 0.015). As the results of a test of the simple main effect, MVC/mCSA in both limbs was significantly increased, but the relative change in MVC/mCSA was higher in ST (almost certain with large) than COMB (very likely with small) (Figure 2).

### 3.3. Time Course of the Measured Variables in the Experimental Period

Figure 3 presents the time course of the measured variables in the experimental period. For muscle size, there was a significant interaction between group and time in MT (F = 2.249, η_p_ = 0.070, *p* = 0.041) and mCSA (F = 2.403, η_p_ = 0.074, *p* = 0.029). Compared to Mid, the significant changes in MT and mCSA were found after three weeks in COMB, but did not change significantly at any time points in ST. For MVC, the ANOVAs revealed that there was a significant main effect of time (F = 35.296, η_p_ = 0.541, *p* < 0.001), but no significant interaction between group and time was found (F = 0.140, η_p_ = 0.005, *p* = 0.955). Both training programs significantly increased MVC after three weeks from Mid. For MVC/mCSA, there was a significant main effect of time (F = 12.567, η_p_ = 0.295, *p* < 0.001), but no significant interaction between group and time was found (F = 1.317, η_p_ = 0.042, *p* = 0.269). Compared to Mid, significant changes in MVC/mCSA were found after three weeks in both arms. The relative change in MVC/mCSA was negatively related to that in mCSA for ST, but the corresponding relationship was non-significant for COMB (Figure 4). Figure 5 illustrates a summary of the current findings obtained for the experimental period.

## 4. Discussion

The main findings obtained here were that, following the first 3-week isometric training to volitional failure, the 6-week isometric training aiming to strengthen MVC led to an increase in MVC, with no change in mCSA, but the training aiming to develop MVC as well as muscle size increased MVC as well as mCSA, although the relative change in MVC was the same in ST and COMB. These findings indicate that the contributions of muscle size and neuromuscular function to strength gain may be different due to training protocols.

In this study, the relative change in mCSA (11.3%) was greater in COMB than ST (0.6%), whereas the strength gain was similar between both arms. The value of COMB is higher than the relative change in muscle cross-sectional area of the knee extensors (approximately 2%) reported by Goto et al. [8]. Since training-induced changes in muscle size are approximately twofold higher in upper limbs than lower limbs [23], it could be considered that the different adaptation of mCSA might be due to the site-related difference. In this study, however, the group-related difference in training-induced change in mCSA was higher than that in the earlier study. Therefore, the increase in mCSA may be attributable to the different training programs in this study. Resistance exercise for developing maximal voluntary strength and muscle hypertrophy, which consists of five repetitions at 90% of one repetition maximum (1RM), as well as a resistance exercise to repetition failure at 30% or 50% of 1RM per set, produces more blood lactate and growth hormones than that for developing maximal voluntary strength alone [24]. They also demonstrated that the resistance exercise with only five repetitions × 90% of 1RM produced less growth hormones [24]. Growth hormone has been clearly shown to stimulate protein synthesis and to facilitate muscle hypertrophy [25]. Further, a single bout of resistance exercise, performed at 30% of 1RM to repetition failure, simulates myofibrillar protein synthesis rates [26]. Considering these findings, the greater increase in mCSA in COMB rather than ST might be affected by the different anabolic hormonal secretion.

Another potential mechanism concerning the different adaptation of muscle hypertrophy is considered to be the group-related difference in mechanical stress in training sessions. Mechanical stress may affect muscle protein synthesis directly [27], via stimulating mammalian target of rapamycin (mTOR) [28] and activating the extracellular regulated kinase/tuberous sclerosis complex 2 pathway [29]. In this study, however, the elbow flexion force in training sessions tended to be lower in COMB than ST (Figure 1) (*p* = 0.073). Therefore, the influence of mechanical stress on the different adaptation of muscle hypertrophy might be less in this study.

Interestingly, the relative change in MVC/mCSA was negatively related to that in mCSA for ST, but the corresponding relationship was non-significant for COMB. Following a 10- and 20-week hypertrophic resistance training regimen, an increase in CSA was shown to be negatively related to changes in muscular activity and voluntary activation in the knee extensors (r = −0.84) [9], indicating that one or the other may predominantly adapt in the adaptation to resistance training for muscle hypertrophy and neural activation. In this study, the increase in mCSA was higher in COMB than ST, so then the current finding is not in agreement with the earlier finding.

The training-induced changes in MVC were of the same extent between COMB and ST. Figure 5 illustrates a summary of the current findings obtained during the experimental period. In earlier studies [5,30], the contribution of muscle hypertrophy to maximal voluntary strength has been discussed. Buckner et al. [30] revealed that resistance training with a load of 1RM induced a similar increase in maximal strength as resistance training with the traditional training regime (70% 1RM), suggesting that muscle hypertrophy may not increase the potential for strength gain. In this study, however, the relative change in MVC/mCSA for COMB was lower than that for ST, from Mid to Post. Further, MVC/mCSA at Post tended to be lower in COMB than ST, with small (0.52), although no significant difference in MVC/mCSA between ST and COMB at Mid was found, with trivial (0.18). Maximal voluntary strength per muscle cross-sectional area is constant in the elbow flexors, regardless of age, sex, and training background [14,31]. Combining the current findings with the earlier findings, muscle hypertrophy in COMB will allow the potential for strength gain through subsequent resistance training of the ST-type.

In this study, different training programs were prescribed to the left and right arms of a participant. Hence, we should state an effect of cross-education on the strength gain. To examine the effect, we computed the correlation coefficient between strength gains in ST and COMB arms. The coefficient was 0.48, which was not significant. Further, the strength of the corresponding relationship was small. In previous studies, strength gain in an untrained arm in unilateral strength training was less than 11% [32,33], lower than the strength gain (22–25%) in this study. Considering these findings, it is possible that the influence of cross-education on maximal voluntary contraction might be small.

## 5. Conclusions

Following a 3-week isometric training to volitional failure (60–80% MVC), a 6-week training regimen for developing maximal voluntary and muscle hypertrophy increased MVC with increasing mCSA, and the training-induced change in MVC was similar to that for developing maximal voluntary strength alone. The current findings suggest that, following resistance training as familiarization, a high-intensity training regimen, adding low-intensity with volitional failure, leads to an improvement of maximal voluntary strength as well as muscle hypertrophy simultaneously. Based on the time courses of mCSA and MVC in this study, on the other hand, strength conditioners need to be careful about keeping that training going for more than three weeks.

## Figures and Tables

**Figure 1 ijerph-20-03837-f001:**
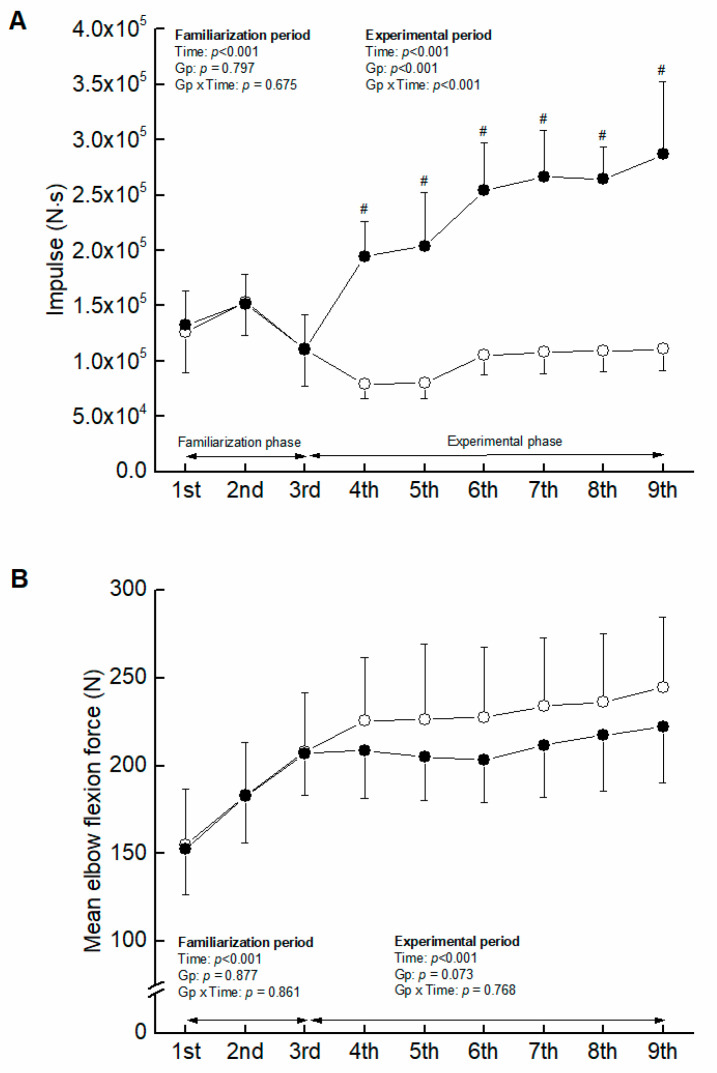
Comparison of impulse (**A**) and mean elbow flexion force exerted (**B**) between ST and COMB in training sessions. The filled and open circles denote COMB and ST, respectively. #, significant group-related difference.

**Figure 2 ijerph-20-03837-f002:**
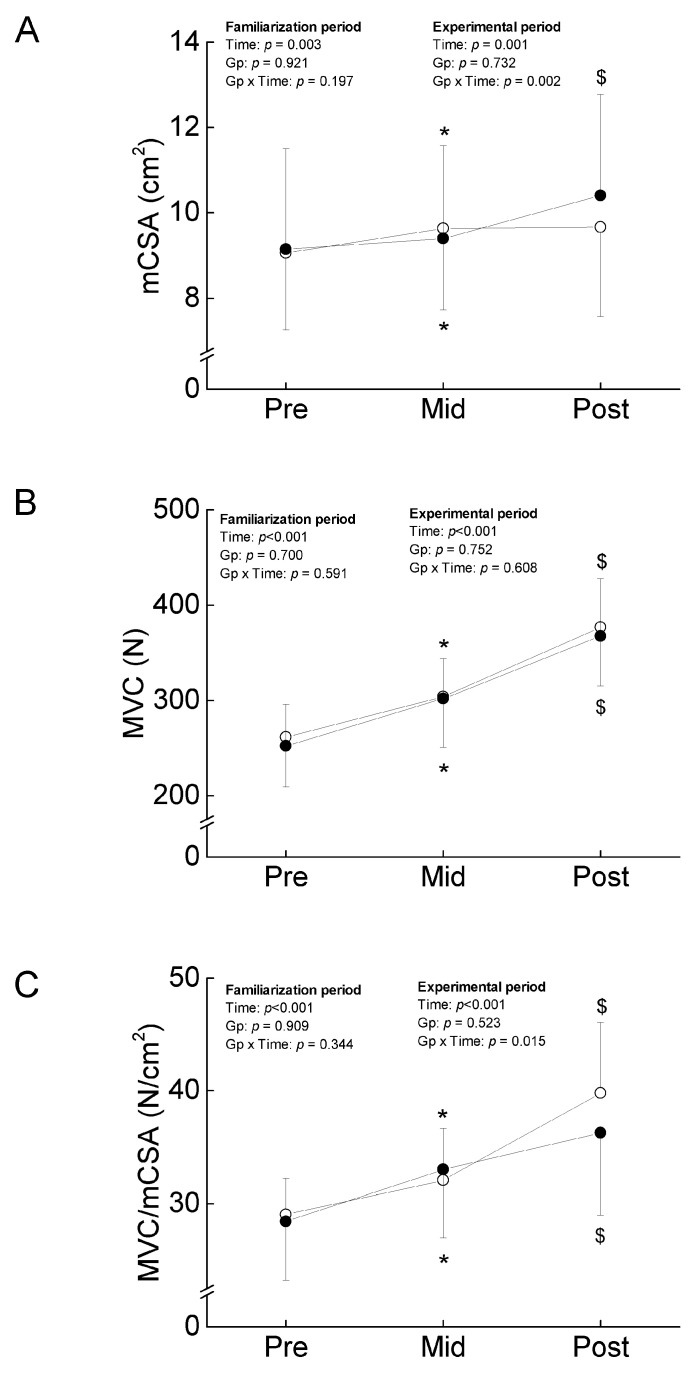
Training-induced changes in muscle cross-sectional area (mCSA) (**A**), maximal voluntary isometric contraction (MVC) (**B**), and MVC/mCSA (**C**) of the elbow flexors through the intervention. The filled and open circles denote COMB and ST, respectively. *, significant difference compared to Pre; $, significant difference compared to Mid.

**Figure 3 ijerph-20-03837-f003:**
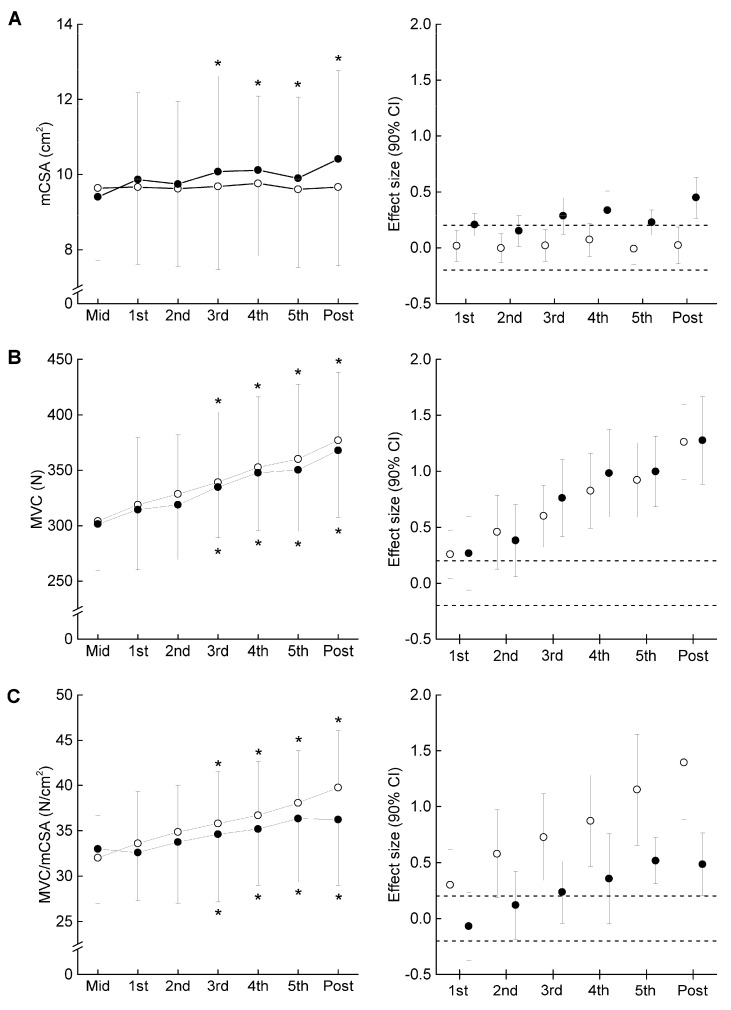
Time course of muscle cross-sectional area (mCSA) (**A**), maximal voluntary isometric contraction (MVC) (**B**), and MVC/mCSA (**C**) in the elbow flexors from Mid to Post (left panel). Effect size with 90% confidential interval (CI) at each measurement point compared to Mid (right panel). The filled and open circles denote COMB and ST. *, significant difference within the same arm compared to Mid. Dotted line is presented as ± 0.2 of effect size.

**Figure 4 ijerph-20-03837-f004:**
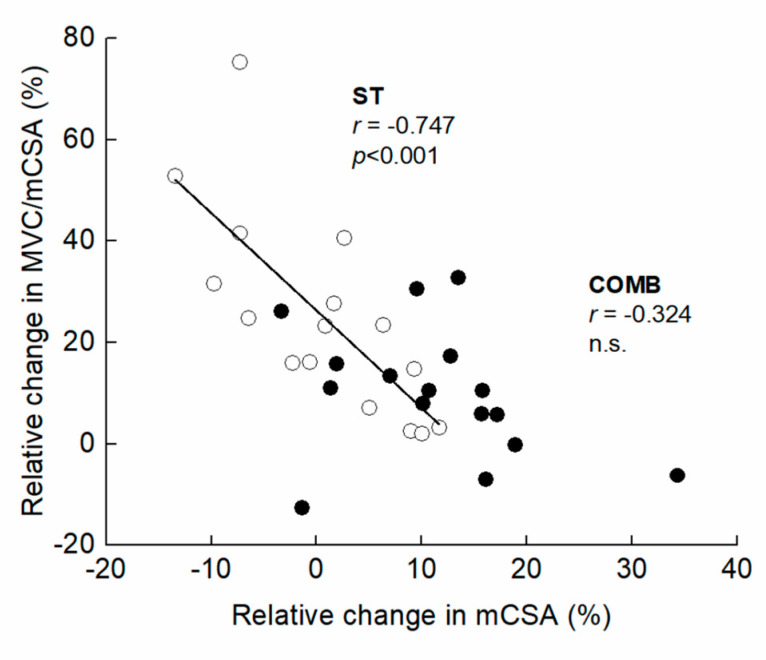
Association of relative change in maximal voluntary isometric contraction per muscle cross-sectional area (MVC/mCSA) with relative change in mCSA from Mid to Post for the elbow flexors. The filled and open circles denote COMB and ST, respectively.

**Figure 5 ijerph-20-03837-f005:**
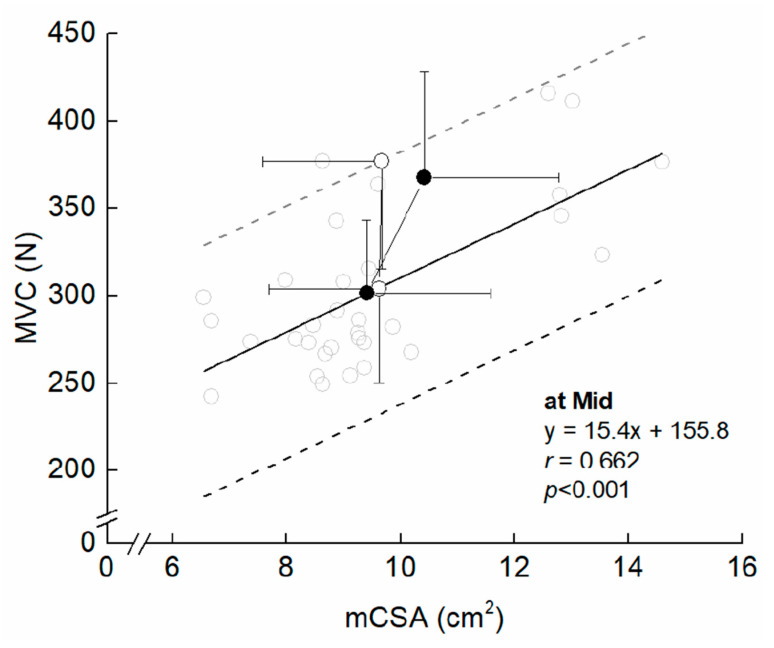
Training-induced changes in maximal voluntary isometric contraction (MVC) and muscle cross-sectional area (mCSA) in the elbow flexors. The grey opened circle denotes an individual’s plot at Mid. The black filled circles indicate means and SDs in COMB, from Mid to Post. The open circles represent means and SDs in ST, from Mid to Post. The black line indicates regression line derived from the corresponding relationship. The dashed line indicates ± 2SDs derived from the single regression line.

**Table 1 ijerph-20-03837-t001:** Training schemes in ST and COMB.

	Familiarization Period			Experimental Period					
Week	1st	2nd	3rd	4th	5th	6th	7th	8th	9th
ST	60	70	80	90	90	90	90	90	90
	4	4	4	3	3	4	4	4	4
	TF	TF	TF	5	5	5	5	5	5
	1	1	1	3	3	3	3	3	3
COMB	60	70	80	90 + 50	90 + 50	90 + 50	90 + 50	90 + 50	90 + 50
	4	4	4	3	3	4	4	4	4
	TF	TF	TF	5 + 1	5 + 1	5 + 1	5 + 1	5 + 1	5 + 1
	1	1	1	3 + TF	3 + TF	3 + TF	3 + TF	3 + TF	3 + TF

ST: a group aiming to strengthen maximal voluntary strength. COMB: a group aiming to develop muscle size as well as maximal voluntary strength. TF: task failure. Row 1: Training intensity (%MVC). Row 2: Number of set (times). Row 3: Number of contractions per set (times). Row 4: Contraction time (s) in each group.

## Data Availability

We will provide for dataset after publication.

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
