# Peer review of "Effects of Different Isometric Training Programs on Muscle Size and Function in the Elbow Flexors"

_ijerph, 2023, doi:10.3390/ijerph20053837_

Round 1
Reviewer 1 Report
Dear authors
Congratulations for the manusccript. The topic is of great interest and the methodology appears to be OK. In order to be published in the Journal, please consider the following comments.
Abstract: please explain all the abbreviations the first time they appear in the text
Line 118: please consider to explain and describe the procedures for obtaining the images. It is interesting to repeat the measurements of a previous article but in my opinion a brief explanation is important in this section.
Figures: some sentences are difficult to read on the figures. Please improve the quality since some information appears to be missing. especially in fig 1
Figure 5 and its explanation are suggested to change their location to the results section. This information is useful in results section and authors could refer to it in the discussion section, but no figures or results are correct in the discussion.
References: Please review this section according to the journal guidelines
Reviewer 2 Report
Congratulations to the authors for bringing to light the prescription and control of isometric training.
It is important that the authors describe in the methods how the isometric training loads were (started with how many % of CVIM and ended with how many %).
This is fundamental to understand the adaptations described in the results, discussion and fundamental to support the conclusions.
Reviewer 3 Report
There seems to be no recognition of potential for cross-education. Cross-education has been mostly covered as a consequence of unilateral training. Do the authors think that cross-education for the strength measurements was not possible. This could be a serious limitation/weakness of the study and needs to be discussed.
L11. “It remains questionable”. Maybe better to replace with “It remains unknown”.
Ls 23-24. The conclusion seems to be a statement of results. I suggest to revise.
L81. Please provide the target force.
Table 1 is not very clear. I suggest to provide the familiarization for both groups because then the mention of rows in the legend is clearer.
Ls 103-112. Move this to the start of the methods section as it is e.g. stated in L140 “the following intervention”.
L104. I suggest to change “22.4 ± 2.5 years, 174.1 ± 6.5 cm, 75.8 ± 14.1 kg)” to “22 ± 3 years, 174 ± 7 cm, 76 ± 14 kg)”.
L153. Please revise “Data was presented descriptive data as mean ± SD.”.
L166. What is meant by the corresponding phase?
I suggest to label the panels in the figures (a etc) and change the figure legends.
L267. I suggest to revise “but the significance level did not reach”.
L276. Change “is not support” to “is not supported”.
L291. Please reconsider/revise/clarify “Unfortunately, we had the relevant data, and further investigation is needed in this point”.
Ls 310-314. I suggest to add a conclusion that could be useful for strength and conditioning trainers and coaches and athletes.
Round 2
Reviewer 2 Report
Congratulations to the authors for complying with the suggested chang
Author Response
We have appreciated reviewing our proof.
Your comments have made our proof better.

Reviewer 3 Report
L24. mCSA is used for the first time here. Please define.
Author Response
We have appreciated reviewing our proof.
Your comments have made our proof better.
Comment #1
L24. mCSA is used for the first time here. Please define.
Response to comment #1
We have done. We have added the following sentence to abstract.
" Muscle cross-sectional area (mCSA) was derived from the obtained muscle thickness. "
